# Can Physiological and Psychological Factors Predict Dropout from Intense 10-Day Winter Military Survival Training?

**DOI:** 10.3390/ijerph17239064

**Published:** 2020-12-04

**Authors:** Jani P Vaara, Liisa Eränen, Tommi Ojanen, Kai Pihlainen, Tarja Nykänen, Kari Kallinen, Risto Heikkinen, Heikki Kyröläinen

**Affiliations:** 1Department of Leadership and Military Pedagogy, National Defence University, P.O. Box 7, 00861 Helsinki, Finland; heikki.kyrolainen@jyu.fi; 2Finnish Defence Research Agency, Finnish Defence Forces, P.O. Box 5, 04401 Järvenpää, Finland; liisa.eranen@mil.fi (L.E.); tommi.ojanen@mil.fi (T.O.); kari.kallinen@mil.fi (K.K.); 3Personnel Division of Defence Command, P.O. Box 919, 00130 Helsinki, Finland; kai.pihlainen@mil.fi; 4Army Academy, 53600 Lappeenranta, Finland; tarja.nykanen@mil.fi; 5Statistical Analysis Services, Analyysitoimisto Statisti Oy, 40720 Jyväskylä, Finland; risto.heikkinen@statisti.fi; 6Faculty of Sport and Health Sciences, University of Jyväskylä, P.O. Box 35, 40114 Jyväskylä, Finland

**Keywords:** soldiers, attrition, physical fitness, winter, cold environment, resilience, combat readiness

## Abstract

*Background:* In the military context, high levels of physiological and psychological stress together can compromise individual’s ability to complete given duty or mission and increase dropout rates. The purpose of this study was to investigate if baseline physical fitness, body composition, hormonal and psychological factors could predict dropout from a 10-day intense winter military survival training. *Methods:* 69 conscripts volunteered to participate in the study. Physical fitness (muscle strength and power, muscle endurance, and aerobic fitness), body composition and hormonal variables (BDNF, testosterone, cortisol, SHBG, DHEAS, IGF-1) together with self-reported psychological factors (short five personality, hardiness, sense of coherence, stress, depression) were assessed prior the survival training. *Results:* During the survival training, 20 conscripts (29%) dropped out. Baseline aerobic fitness (hazard ratio, HR: 0.997, 95% CI: 0.994–0.999, *p* = 0.006) and serum cortisol (HR: 1.0006, 95% CI: 1.001–1.011, *p* = 0.017) predicted dropout in Cox regression model. Each 10 m increase in the 12 min running test decreased the risk for dropout by 3%. *Conclusion:* Although most of the physiological and psychological variables at the baseline did not predict dropout during a short-term winter survival military training, baseline information of aerobic fitness and serum cortisol concentration may be useful to target support for individuals at higher potential risk for dropout.

## 1. Introduction

High levels of physiological and psychological stress are often encountered by soldiers during sustained military operations. Occupational stressors including sleep deprivation, prolonged physical activity, energy and fluid deficit together can compromise individual’s ability to complete given military duty or mission [1]. These high physiological and psychological demands can even lead an individual to voluntarily give up or force to dropout from arduous military training, which in case of operation could have deleterious effects on mission success for the whole unit. Most of the previous military studies investigating dropout or attrition have focused on attrition from military service (e.g., [2]) or specific Special Force courses (e.g., [3]). Nevertheless, less is known about predictors associated with dropout during an intense short-term military field training. During military service, field training serves as one of the most important practical educational settings to practice relevant soldier skills where basics learned during garrison training are applied into operative scenario under higher physiological and psychological stress. Therefore, dropout from the military field training may compromise individual learning results and achievement of educational goals for the military unit.

Especially important in the field training is the fact that physiological and psychological stress can be simulated to meet real-life scenarios regarding military missions. Therefore, field training often includes high demands for both psychological and physiological tolerance due to energy deficit, sleep restriction and high amount of physical activity combined with demands for executive function [4,5,6,7]. Previous studies have reported negative changes in physical fitness, body composition and hormonal profile after intense military field training periods [4,5] as well as after survival field training [8,9,10]. Earlier studies focusing on dropout from military service and a Special Forces course [3,11] have revealed that the level of physical fitness may be associated with dropout. Saxon et al. (2020) observed that there was a tendency for higher pull-up and 3-mile run test performances, but not in sit-ups, in individuals who were able to finish 25-day Reconnaissance course compared to drop-outs [3]. Another study by Gottlieb et al. (2020) showed that a modified loaded submaximal endurance test was inversely associated with dropout during 18 months infantry training in women, whereas upper limb stability test and BMI were not [11]. However, there are appears to be lack of studies investigating dropout in a short-term intensive military survival training performed in demanding winter field conditions. Moreover, hormonal biomarkers respond rapidly to alterations in physiological state and it has been shown that during military field training, for example, cortisol and testosterone concentrations can change to great extent [4,5,8,9,11]. Nevertheless, interactions between baseline concentrations of hormonal profile and dropout from military survival training has not been previously studied.

In addition to high physiological burden often involved in the military field training, psychological demands may be increased and challenged [6,12]. The field training typically include energy and sleep deficit and they further expose individuals to negative changes in mood and may further set challenges to mental executive function [6,7]. It has been shown that affective reactivity to stressors vary based on sleep duration such that sleep loss may amplify loss of positive affect on days with stressors [13]. Together, energy and sleep deficit may compromise critical psychological and cognitive abilities during the field training [6,7]. Although no previous studies have assessed whether psychological parameters are associated with dropout in intense military field training, some of the previous studies have focused on military service or shorter specialised military courses [3,14]. Generally, the most attrition occurs in the beginning of the military service [14] and discharge from mental reasons are one of the most common reasons. In a shorter Special Forces course, extraversion and positive affects personality traits were associated with lesser likelihood for attrition from 25-day Reconnaissance course [3], suggesting a possible link with psychological measures and dropout phenomena. Hardiness, conscientiousness, emotion regulation and self-efficacy are known to be related with better functioning and endurance in challenging situations. Hardiness is a personality trait marked by a high level of commitment, control and challenge and it has been associated with good health and high performance in both civilian and military samples [12]. Previous studies have indicated that five factor personality measures are valid predictors of job performance and success in high-risk occupations. Conscientiousness and Emotional Stability seem to predict success in military training programs [15]. Bandura (1997) has shown that self-efficacy plays a central role in the acquisition and maintenance of adaptive behavior patterns [16] and Souza et al. (2014) studied this connection in a sample of military cadets [17]. Self-efficacy acts as a buffer in a threatening situation as the perception of a situation will depend on the way the individual interprets it and on his coping skills. Moreover, hardiness has been shown to protect from stress symptoms in military related stress [18,19,20]. To the best of our knowledge, associations of hardiness, conscientiousness, emotion regulation and self-efficacy with dropout in intensive military survival training has not been widely studied.

The main aim of the present study was to investigate if baseline physical fitness, body composition, hormonal and psychological factors could predict dropout from a 10-day intense winter military survival training. It was hypothesized that some of the physical fitness (aerobic fitness and strength) and psychological variables (hardiness) may predict dropout. To the best of our knowledge, there are no studies that have addressed factors predicting dropout during a short-term arduous winter military survival training.

## 2. Materials and Methods

### 2.1. Military Winter Survival Training

A winter military survival training is an integral part of training of conscripts in the northern parts of Finland during their compulsory military service. The participants’ mean age was 19 ± 1.0 and the majority of the participants exercised at least 2–5 h a week (46%) or more than 5 h a week (48%). The selection criteria for the sample was based firstly on eligible to participate to the survival training (withdrawal based only on medical doctor’s decision either on sickness or injury) and secondly on voluntariness to participate to the study. All conscripts were from the same unit and all of them were eligible and voluntary to participate to the study. The present winter military survival training lasted 10 days. The days 1–2 included theoretical and practical military training education and preparatory actions for the field training. The days 3–5 included transitions and survival training in the field conditions. In total, 25 out of 69 participants were randomized prior the survival training into “recovery intervention” for the days 7 and 8 to an indoor accommodation, where they could rest and eat and drink ad libitum, take part in relaxation exercises, yoga and mindfulness sessions between intense field training. Four out of these 25 participants dropped out before intervention, leaving the total number of participants that took part of the intervention 21 (Figure 1). The subjects of the recovery intervention participated in yoga and mindfulness sessions that lasted about an hour and took place alternating twice a day. In addition, relaxing background music (60 bpm) was played during leisure time. This group returned from the “recovery intervention” to the field training on day 9. The rest of the study sample (*n* = 34) continued their field training from day 6 onwards by military survival training until the end (day 10) of the field training (Figure 1). The observations of the participants who were randomized to the intervention were intentionally censored after 6 days to exclude the confounding intervention effect on drop out. Therefore, the total study sample was 69 from the beginning until day 5 of the field training and thereafter between day 6 and 10 sample size was 44. These numbers do not include the dropouts, which are, however, shown in the Figure 1 and in the results. During the last day (day 10), participants returned back to garrison from the field conditions (baseline characteristics of the subjects are presented in Table 1).

The winter survival training included high volume of physical activity, performed mainly by skiing, and combined with energy and sleep deficit. The distance covered by skiing was on average 19.3 ± 1.7 km/day during days 3–5 for the whole study group and 13.8 ± 1.3 km/day for those participants who remained in the field conditions (*n* = 44) during days 6–9. Each participant carried around 35–40 kg of external load during the field training. The amount of sleep, energy expenditure, and mental and physical demands during the military survival training are described in Figure 2. The amount of sleep was self-reported daily. Energy expenditure was based on heart rate that was monitored continuously during the training (Firstbeat Ltd., Jyväskylä, Finland) and are expressed for only those days including measurement for the whole day (day 1 and 10 excluded). Mental and physical demands were self-reported assessed by the NASA Task Load Index (NASA TLX) questionnaire using 0–20 scale. In the Figure 2, the results among the whole study group (*n* = 69) are shown during days 3–5 and during days 6–9 for those participants that remained in the field conditions (*n* = 28). The temperature varied between −11 and +8 °C and the average depth of snow varied between 79 and 101 cm during the field training.

The winter survival training consisted of survival actions in winter environment, such as escape under fire, searching and utilizing food in the forest, maintenance of physical and psychological readiness and coping ability under extreme conditions and building temporary accommodation in the forest. The psychological load included evading an enemy and continuous uncertainty of the upcoming actions during the field training. In addition, continuous sleep restriction and energy deficit negatively affected performance as well.

Baseline physical fitness and psychological measures were assessed 3 days prior the field training, except for 12 min running test, which was assessed during the second week of their military service. Body composition and hormonal biomarkers were assessed one day prior to the field training.

The present study was approved by the Finnish Defence Forces (AO1720) and ethical approval was granted by the Scientific and Ethical Committee of the Helsinki University Hospital Research (HUS/900/2018). All conscripts were informed of the experimental design, and the benefits and possible risks that could be associated with the study prior to signing an informed consent document to voluntarily participate in the study.

### 2.2. Physical Fitness

Maximal isometric force of the upper (bench press) and lower extremities (leg press) were measured bilaterally in a sitting position by an electromechanical dynamometer manufactured by the University of Jyväskylä, Finland. The knee and hip angles were set to 107° and 110° [21] in the horizontal leg press position. Participants were told to keep contact with the seat and the backrest during the performance. For the upper extremities, the equipment was adjusted for each subject to their sitting position with their feet flat on the floor, the arms were parallel to the floor, and the elbow angle was 90°. The test was performed by pushing the bar horizontally. One trial attempt before the two test trials were conducted for both leg press and bench press with a minimum of 60 s for recovery. The participants were instructed to produce maximal force as fast as possible and the testing personnel encouraged them verbally during the maximal effort. The best performances were selected for further analysis.

A 6 s maximal anaerobic power cycle ergometer test (Wattbike Ltd., Nottingham, UK) was used to measure peak power in the lower extremities. The participants were seated stationary at the start with the dominant leg initiating the first downstroke. The weight of the participants was inserted into the test computer and air and magnetic resistance were set according to body weight with variance of options 3–5 (61–95 kg). The test started following a 5 s countdown followed by verbal command [22].

Standing long jump was used to measure explosive force production of the lower extremities [23]. Prior to testing, which was performed on a specifically designed gym mat (Fysioline Co., Tampere, Finland), the participants were instructed of the correct technique, and they performed a warm-up and several practice jumps. The participants were instructed to jump (horizontally) forward as far as possible from a standing position without falling backward upon bilateral landing. The result of the best jump was measured in centimeters as shortest distance from the landing point to the starting line.

Muscular endurance of the trunk and upper extremities was assessed using sit-up and push-up tests (repetitions/minute). Sit-ups were used to assess abdominal and hip flexor performance [24]. and push-ups were used to measure performance of the arm and shoulder extensor muscles [25]. At the start in the sit-up test, a participant laid on his back, while the legs were supported from the ankles by an assistant. The knee angle was 90°, elbows pointing upwards, and fingers crossed behind the head. A successful repetition was counted when the participant lifted his upper body from the starting position and brought elbows to the knee-level. At the start in the push-up test, a participant laid face down on the floor, feet parallel at the pelvis to shoulder width with hands positioned so that the thumbs could reach the shoulders while other fingers pointed forward. Before the test, the participants were instructed to extend their arms to the starting position and keep the feet, trunk, and shoulders in the same line throughout the test. One successful repetition was counted when participant lowered his torso by flexing arms to an elbow angle of 90° and returned to the starting position by extending his arms.

Seated medicine ball throw was used for assessing explosive force production of the upper body. The participants sat upright on the floor with their legs fully extended and back kept against the vertical wall throughout the test. The medicine ball was held with both hands, the forearms positioned parallel to the ground. The participant threw the medicine ball vigorously as far forward as possible while maintaining the back against the wall. The distance from the wall to lnding point of the medicine ball was recorded. The best result out of three throws was used in the analysis. The participants were allowed to have at least three training throws before the measurements.

Aerobic fitness was measured with a 12 min running test [26] on outdoor track in the beginning of the military service (~2 or 8 months prior the beginning of the study depending on each conscript’s service time). Conscripts were encouraged to run with maximal effort at steady pace until the end of the test. The results were recorded as distance ran in 12 min with an accuracy of 10 m.

### 2.3. Body Composition

Body composition was measured in the morning after an overnight fast. Body mass (BM), skeletal muscle mass (SMM) and fat percentage (FAT%) were measured by using the segmental multi-frequency bio-impedance method (BIA) (InBody 720, Biospace Co. Ltd., Seoul, South Korea). Body height was measured by stadiometer in the beginning of their service. Further, body mass index (BMI) was calculated.

### 2.4. Serum Hormone Concentrations

Venous blood samples were drawn from the antecubital vein after an overnight fast 1 day prior to the field training. Serum concentrations of testosterone (TES), cortisol (COR), sex hormone binding globulin (SHBG), insulin-like growth factor-1 (IGF-1) and dehydroepiandrosterone (DHEA) and C-reactive protein (CRP) were analyzed (Siemens Immulite 2000 XPI, Siemens Healthcare, Malvern, PA, USA). The sensitivity and interassay coefficients of variance for these assays were 0.5 nmol/L and 7.8% for TES, 5.5 nmol/L and 6.5% for COR, 0.02 nmol/L and 5.7% for SHBG, 2.6 nmol/L and 7.8% for IGF-1, 0.08 µmol/L and 7.6% for DHEA and 0.1 mg/L and 7.3% for CRP. The samples were centrifuged (Megafire 1.0 R Heraeus, DJB Labcare Ltd., Buckinghamshire, UK) after 30 min at 2000× *g* for 10 min, frozen, and transported to the laboratory for later analysis.

### 2.5. Self-Reported Psychological Measures

Psychological measures were self-reported prior the field training questionnaires including Short Five personality measure [27], hardiness (Dispositional Resilience Scale) [19], Sense of Coherence [28], a short stress scale [29], and a short depression scale [30], as well as cohesion [31]. The Short Five personality measure is a shortened version of the Big Five measuring the five main personality traits: extraversion, agreeableness, neuroticism, openness and conscientiousness. Hardiness (Dispositional Resilience Scale) measures the way a person interprets difficult situations: if he sees them as a problem or as a challenge. In addition, sense of coherence measures how meaningful and comprehensible a person sees life to be. Moreover, a short stress and depression scales were used to measure stress and depression, respectively. Sum score of each questionnaire were used.

Psychological measures of Short Five [27] was used to measure personality traits of extroversion, openness, neuroticism, agreeableness and conscientiousness. The questionnaire consisted of 60 questions with a 7-point Likert scale from −3 (strongly disagree) to +3 (strongly agree).

Hardiness was assessed by the Dispositional Resilience Scale-15 [19] including 15 questions with a Likert scale of 1 to 4 (not at all true, a little true, quite true, completely true) and cohesion with platoon cohesion index with a Likert scale of 1 to 5 [31].

Sense of Coherence Scale [28,32] comprises of three sub-scales measuring feelings of control, meaningfulness and comprehensibility. It measures how much a person feels to have control over his own life, how meaningful life seems to be and how comprehensible the world and other people are to him. The subjects rated questions on a 7-point Likert scale from 1 (strongly disagree) to 7 (strongly agree).

Core Self Efficacy Scale [33]. Self-efficacy is near to self-confidence or self-esteem, but self-efficacy is a feeling of competence related to a certain task or situation. Core Self Efficacy Scale involves four personality traits: locus of control, neuroticism, generalized self-efficacy, and self-esteem. The subjects rated questions on a 5-point Likert scale from 1 (strongly disagree) to 5 (strongly agree).

Depression was assessed with a short (10 items) Likert-type scale consisting of symptoms that a typical for depressive people [30]. The subjects rated questions on a 4-point Likert scale from 1 (strongly disagree) to 4 (strongly agree).

Stress was assessed with a modified short Likert-type scale (11 items) consisting of general stress symptoms [29]. The subjects rated questions on a 5-point Likert scale from 1 (strongly disagree) to 5 (strongly agree).

Adverse Childhood Experiences were also assessed with a questionnaire [34]. It is a widely used scale of childhood experiences and lists many kind of life problems, family violence, illness, death, divorce, sexual abuse, etc., asking which of these a person has experienced in his life and during which life period, pre-school, elementary school, primary school, college. The subjects rated on a yes or no answer if they have experienced these events and if they have, during which age period.

### 2.6. Statistics

Statistical analysis was conducted in R (R Core Team, 2020) with package survival [35]. Data are presented as means with standard deviation and confidence intervals (95% CI) where appropriate. The associations between dropout and the predictor variables (physical fitness, body composition, hormonal biomarkers, psychological measures) were analyzed with several approaches. Although, the main results are related to Cox regressionto assess prediction of all variables of drop out, logistic regressions were also used to describe associations with single variables.Firstly, the robust association between dropout and predictor variables were analyzed with logistic regressions (β-coefficients with 95% confidence intervals) separately for each predictor variable alone (Table 2). This approach does not take into account any information on the time of the occurrence of dropout, but binary variable of dropout was determined based on participants who remained in the survival training throughout the first 9 days. Further, to take into account time point (day) when dropout occurred, Cox proportional hazard ratios were applied. This approach included two steps: Firstly, Cox regressions were analyzed separately among all variables in each subcategory (separately for physical fitness variables, separately for body composition variables, separately for hormonal variables and separately for psychological variables) (Table 3). The model selection was based on a stepwise algorithm (stepAIC function) [36] with both directions and AIC (Aikaike information criteria) as a goodness-of-fit criteria. Later, this algorithm is called STEP. Secondly, the predictors from all subcategories that “passed” STEP algorithm in the first stage were included in the final model in Cox regressions using STEP algorithm (Table 4). This study is part of the bigger study project, where 25 participants started recovery intervention from day 6 onwards and, therefore, no dropout among them were possible, whereas other remaining participants continued their field training. Due to this study design, the observations of the participants who were randomized to the intervention were intentionally censored after 6 days to exclude the confounding intervention effect on drop out. In addition, group information (recovery intervention, control = continuous field training group) were used as a covariate in all of the regression models used. The level of significance was set at <0.05.

## 3. Results

During the winter survival military field training, 29% (*n* = 20) of the conscripts dropped out. The mean day when dropout occurred was 3.9 ± 1.0 during the first 5 days (*n* = 14) when the whole study sample was included (*n* = 69), and for the remaining study sample (*n* = 34) thereafter (days 6 onwards) 7.5 ± 1.4 (*n* = 6). Demographic, physical fitness, body composition, hormonal and psychological factors for dropouts and those completing the survival training are presented in Table 1.

The prediction model for robust dropout (not taken into account at what day dropout happened) using logistic regressions separately for each variable alone showed that baseline 12 min running test (β: −0.003, 95% CI: −0.006–−0.0002, *p* = 0.047) and sit-up performance (β: −0.09, 95% CI: −0.18–−0.02, *p* = 0.025) were inversely associated with dropout (Table 2). No statistically significant associations of body composition and psychological factors with dropout were observed (Table 2). Only DHEAS was positively associated with dropout among hormonal variables (β: 0.15, 95% CI: 0.01–0.31, *p* = 0.042) (Table 2).

When Cox regressions were applied separately to subgroup of physical fitness parameters, 12 min running test was borderline significant (hazard ratio, HR: 0.998, 95% CI: 0.996–1.00002, *p* = 0.052) (Table 3). When Cox regressions were applied separately to subgroup of body composition, none of the body composition variables were associated with drop out (Table 3). When Cox regressions were applied separately to subgroup of hormonal variables, cortisol predicted dropout (HR: 1.006, 95% CI: 1.0003–1.01, *p* = 0.039). None of the psychological variables were significantly associated with dropout, but when Cox regressions were applied separately to subgroup of all psychological variables, self-reported depression scale was close to significant (HR: 0.10, 95% CI: 0.009–1.17, *p* = 0.066) (Table 3).

Cox regressions (proportional hazard ratios) revealed that baseline aerobic fitness predicted dropout in the final model (hazard ratio, HR: 0.997, 95% CI: 0.994–0.999, *p* = 0.006), meaning that each 10 m increase in a 12 min running test decrease the risk for dropout by 3% (Table 4). In the same model, baseline cortisol (HR: 1.0006, 95% CI: 1.001–1.011, *p* = 0.017) predicted dropout, whereas IGF-1 (HR: 1.082, 95% CI: 0.994–1.177, *p* = 0.069) was borderline significant (Table 4). Each 1 nmol/L increase in serum cortisol increase the risk for dropout by 0.06%

## 4. Discussion

The main findings revealed that among physical fitness, body composition, hormonal biomarkers and psychological factors baseline aerobic fitness, cortisol and IGF-1 concentrations predicted dropout from an intense winter survival military field training consisting of high physiological and psychological load with energy and sleep deficit. The most prominent prediction for dropout were evident for aerobic fitness and cortisol showing statistical significance. The results indicate that aerobic fitness and cortisol levels could be taken into consideration to target support for those individuals at higher potential risk for dropout during the field training.

Among the widely measured physical fitness components in the present study, aerobic fitness predicted dropout, whereas strength or power measures had no prediction in the final Cox regression model. A plausible explanation may be that military field training lasting several days, including high volume of physical activity combined with energy and sleep deficit, requires especially aerobic fitness to endure these demands. Therefore, conscripts with lower aerobic fitness may have reached the level of physical and/or mental exhaustion well before those with higher fitness, and the exhaustion may have exposed them for drop out. This is important given that in addition to better stress tolerance, higher aerobic fitness has been associated with improved ability to maintain cognitive performance within the military context [37]. Strength and power do not have similar predictive power to higher risk for dropout, probably in part, because these fitness components do not similarly prevent accumulated fatigue during military field training than aerobic fitness. It must, however, be emphasized that strength and power, although not predictive of drop-out in the present study, are important in various military tasks, such as manual material handling and load carriage [38]. Higher level of strength and power may also improve movement economy and thereby, delay fatigue during prolonged submaximal military work [39] and decrease risk for injuries [40]. Therefore, impact of strength and power can be seen additive to aerobic fitness and muscular endurance in regards of soldiers’ overall combat readiness [38] but not for drop-out. Although no previous studies exist investigating prediction of physical fitness with dropout in a military field training, Saxon et al. (2020) studied dropout during a 25-day Reconnaissance Special Forces course [3]. They found a tendency for higher pull-up and 3-mile run test performances in individuals who were able to finish the 25-day Reconnaissance Marine course compared to dropouts [3]. Nevertheless, no association of sit-up performance was demonstrated. A limitation to study by Saxon et al. (2020) is that only statistics of group comparisons were conducted without more sophisticated statistical methods such as regressions, and thus relationships with adjusted confounders were not reported. Another study by Gottlieb et al. (2020) found that a modified loaded submaximal endurance test was associated with attrition for 18 months of infantry training in women, whereas upper limb stability test was not [11]. The present study findings together with the very few recent study results [3,11] may indicate that selected physical fitness components are related to dropout phenomenon, showing most promise in aerobic fitness. However, future studies are warranted to gain more insight in this particular research issue in military settings.

Serum cortisol level assessed at rest prior the military field training predicted dropout in the present study. The cortisol hormone represents catabolic properties and therefore its increase can be reflective of increased stress, whether physical or psychological or both, creating catabolic milieu. As baseline cortisol concentration predicted drop-out, it may be speculated that increased levels could be a result of intensive physical training load [41], mental anticipation prior the field training or their combination. If higher cortisol levels reflect a sign of overreaching or overtraining, then it could be speculated that demanding field training would be harder for dropouts to tolerate because of accumulated fatigue already evident prior the field training. This again would predispose them to increased risk for drop out. Moreover, intensive physical training has been suggested to be related to immune function [42]. If higher levels of cortisol concentration prior the field training were also reflective of overreaching or overtraining and thereby diminished immune function, it may well be suggested that it would also predispose increased risk for drop-out. On the other hand, C-reactive protein at baseline was not associated with dropout, which indicates that at least in baseline no severe deficit in immunology was observed based on C-reactive protein measure.

Even though anthropometrics and body composition are associated with physical fitness and occupational military performance [38], body composition variables were not associated or predictive with dropout in the current study. In line with the present study, Gottlieb et al. (2020) observed that BMI was not associated with attrition during an 18-month infantry training period in women [11]. It may appear that narrow distribution of body composition variables within the study sample may have affected the results towards non-significant findings. Furthermore, it may be that the ones with high body fat percentage or low skeletal muscle mass, both regarded as negative for soldiers’ physical performance, may have had good aerobic fitness and therefore this masks predictive capacity with body composition variables, as have been shown previously in US enlistees [43].

Although we observed no associations between psychological measures and dropout in the present study, there was a slight tendency for depressive thoughts to be positively associated (*p* = 0.066) in Cox regression model including all psychological variables. Nevertheless, in Cox regressions including all the measured physiological and psychological variables none of the psychological metrics were predictive of drop out. In contrast to the present study findings, Saxon et al. (2020) demonstrated that among the Big Five personality traits, extroversion and positive effects were associated with lesser likelihood for dropout from a 25-day Special Forces Reconnaissance course [3]. Nevertheless, in line with the present study, where hardiness did not predict drop out, grit scale used in the study of Saxon et al. (2020) did not predict drop out. It may be that hardiness at baseline is not predictive of drop out, whereas the change in hardiness during the military field training may be more sensitive marker, as suggested by a study of Eid and Morgan (2006), who observed that hardiness was associated with peritraumatic dissociation in a simulated prisoner of war exercise in cadets [12]. The partly controversial results between the present and Saxon et al. (2020) study may originate from different study population being conscript in the present study and Special Forces operators in Saxon et al. (2020) [3].

Interestingly, a recent study showed that stress mind-set predicted success in a 7-week NAVY SEALS training, known to be physically and mentally very challenging [44]. “Stress is enhancing” mindset was inversely related to dropout and positively to obstacle course time [44]. It is worth noting, that all the previous studies investigating psychological metrics and drop-out have used longer exposure time frame (from 25 days up to several months) compared to shorter one used in the present study. The longer exposure time frame may emphasize the magnitude of baseline psychological variables more than the one in the present study, where the military field training lasted only 10 days.

Although, dropout during military field training has not been previously studied, several studies have shown that attrition from psychological measures have been associated with attrition in military service. In fact, psychological symptoms are the most common reason for attrition from basic training period [45,46]. It is, however, important to distinguish between entries in military service and the present study sample. The conscripts of the present study had continued their service after the first weeks of the service, known as the time when most of the attrition occurs [47]. In fact, the majority of the conscripts in the present study were in the latter part of their service (at 9 months) period lasting 12 months in total. Therefore, it may be speculated that they had mentally adapted to strenuous military training, including intense winter military survival training and therefore, psychological measures at baseline prior the field training were not predictive of drop-out. In line with this theory, previous studies have shown that adaptation to psychological distress occurs during basic training period [47,48]. Although psychological metrics did not show prediction of drop out, in general, enhancing mental skills have shown to be beneficial for soldiers in order to improve cognition and stress control during stressful environment, such as the field training [49].

## 5. Strengths and Limitations

Wide variety of physiological and psychological measures were used in the present study. Among physical fitness, all components including aerobic fitness, muscular endurance, maximal strength and power as well as short-term anaerobic fitness were measured. In addition, an extensive hormonal profile, including the most relevant hormones according to previous studies [4,5] was analyzed. Psychological measures were extensive as well, consisting of components of short five personality, hardiness, sense of coherence, stress, depression. Nevertheless, many of the psychological aspects could not be assessed, given the resources available. The present study includes some limitations. The motivation towards the field training was not assessed which can be seen as a limitation. It is presumable that highly motivated individuals tended to overcome the physical and mental stress of intense training and accumulation of fatigue, including energy and sleep deficit better than those with lower motivation. In addition, due to small sample size it was not possible to add testing of out of sample performance of the final prediction model.

## 6. Conclusions

Most of the physiological and psychological variables at baseline did not predict dropout in a 10-day arduous winter military survival training. Nevertheless, aerobic fitness and serum cortisol concentration showed the most prominent and statistical significance to predict dropout, while no other components of physical fitness or other hormonal biomarkers were associated with the drop out. Neither body composition nor psychological variables showed any associations with drop out. The results indicate that although dropout most likely is a complex phenomenon, aerobic fitness and cortisol levels could be taken into consideration to target support for those individuals at higher potential risk for dropout during arduous winter military field training. The findings may indicate that baseline information of these variables may be beneficial in order to target support for those individuals at higher potential risk for dropout during the field training.

## Figures and Tables

**Figure 1 ijerph-17-09064-f001:**
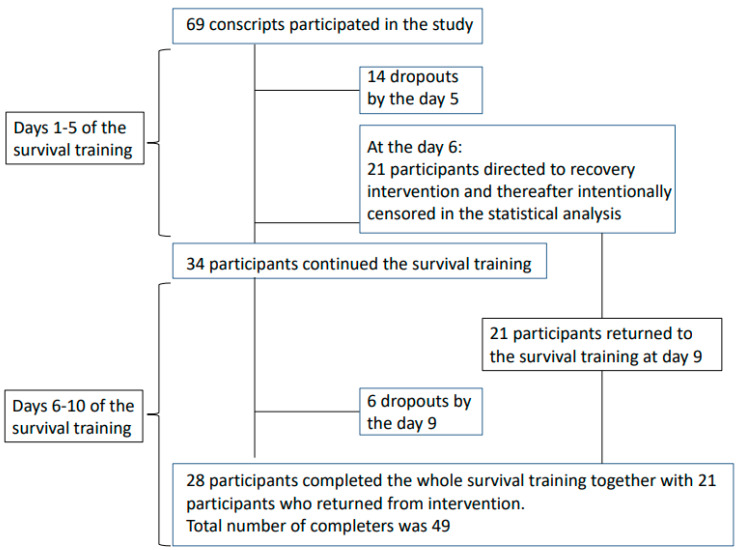
Flow chart of the participants during the military survival training.

**Figure 2 ijerph-17-09064-f002:**
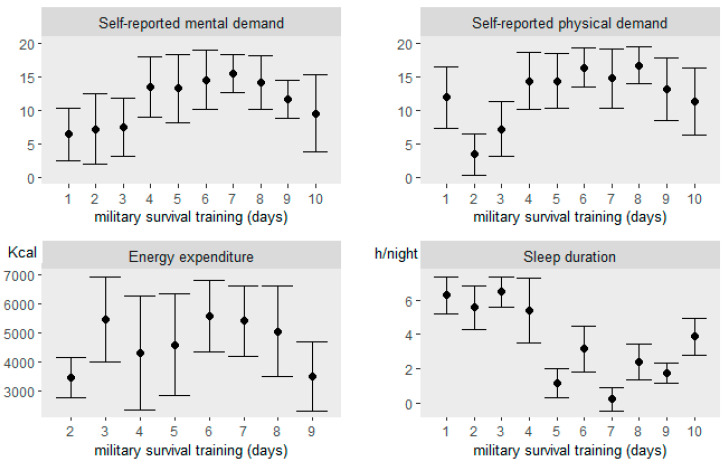
Self-reported physical demand and mental demand assessed by NASA Task Load Index (NASA TLX), and sleep duration, and energy expenditure during military survival training.

**Table 1 ijerph-17-09064-t001:** Demographic, physical fitness, body composition, hormonal and psychological factors in those completing survival training and those who drop-out (mean + sd).

Characteristics	Completers (*n* = 49)	Drop Outs (*n* = 20)
Age (yrs.)	19.6 ± 1.0	19.6 ± 0.8
**Physical fitness**		
12 min running test (m)	2754 ± 196	2652 ± 247
Sit-ups (reps/min)	43 ± 8	39 ± 9
Push-ups (reps/min)	37 ± 12	33 ± 15
Standing long jump (cm)	224 ± 19	214 ± 23
Maximal isometric force of the lower extremities (N)	3230 ± 810	2840 ± 860
Maximal isometric force of the upper extremities (N)	870 ± 210	800 ± 170
Medicine ball throw (cm)	603 ± 78	592 ± 69
6 s maximal cycle performance (max W)	837 ± 131	771 ± 132
**Body composition**		
Body mass index	23.2 ± 2.6	22.5 ± 3.0
%Body fat	13.1 ± 4.9	13.8 ± 6.1
Body mass (kg)	75.0 ± 10.4	72.3 ± 11.1
Body height (cm)	179.7 ± 6.7	179.2 ± 5.7
Skeletal muscle mass (kg)	36.9 ± 4.9	35.1 ± 4.2
Fat mass (kg)	10.0 ± 4.7	10.4 ± 6.2
**Hormonal profile**		
Testosterone (nmol/L)	17.5 ± 6.0	20.1 ± 5.8
DHEA (nmol/L)	8.65 ± 3.93	10.71 ± 4.17
Cortisol (nmol/L)	439 ± 103	476 ± 115
BDNF (nmol/L)	19.0 ± 5.5	18.1 ± 5.5
CRP (nmol/L)	1.77 ± 3.64	1.73 ± 2.86
IGF-1 (nmol/L)	25.2 ± 4.5	27.0 ± 5.2
SHBG (nmol/L)	29.6 ± 11.5	27.1 ± 9.2
**Psychological factors**		
Hardiness	2.99 ± 0.34	2.87 ± 0.36
Sense of Coherence	4.87 ± 0.74	4.76 ± 0.68
Depression	1.38 ± 0.36	1.48 ± 0.52
Stress	1.91 ± 0.47	2.05 ± 0.54
Cohesion	3.82 ± 0.38	3.74 ± 0.48
Big Five: neuroticism	−1.45 ± 0.92	−1.35 ± 0.95
Big Five: extraversion	0.96 ± 0.98	0.70 ± 1.00
Big Five: openness	0.90 ± 0.95	0.60 ± 0.85
Big Five: agreeableness	1.32 ± 0.80	1.18 ± 0.87
Big Five: conscientiousness	1.48 ± 0.88	1.27 ± 0.76
Adverse childhood experiences	30.12 ± 2.06	29.95 ± 2.28
Core self-evaluation: self-esteem	2.38 ± 0.47	2.48 ± 0.42
Core self-evaluation: generalized self-efficacy	2.65 ± 0.29	2.70 ± 0.26
Core self-evaluation: locus of control	2.99 ± 0.60	2.93 ± 0.55
Core self-evaluation: neuroticism	3.63 ± 0.86	3.63 ± 0.90

**Table 2 ijerph-17-09064-t002:** The associations (logistic regressions) of baseline physical fitness, body composition, hormonal and psychological factors with drop-out from the survival training.

Characteristics	Beta-Coefficients + 95% CI	*p*-Value
**Physical fitness**		
12 min running test (m)	−0.003 (−0.006–−0.0002)	0.047
Sit-ups (reps/min)	−0.09 (−0.18–−0.02)	0.025
Push-ups (reps/min)	−0.03 (−0.08–0.009)	0.139
Standing long jump (cm)	−0.03 (−0.05–−0.008)	0.054
Maximal isometric force of the lower extremities (kg)	−0.06 (−0.01–0.0006)	0.090
Maximal isometric force of the upper extremities (kg)	−0.02 (−0.05–0.006)	0.144
Medicine ball throw	−0.002 (−0.01–−0.005)	0.510
6 s maximal cycle performance (max W)	−0.003 (−0.008–0.001)	0.164
**Body composition**		
%Body fat	0.02 (−0.08–0.13)	0.641
Body mass (kg)	−0.02 (−0.08–0.03)	0.493
Body height (cm)	−0.001 (−0.09–0.08)	0.432
Skeletal muscle mass (kg)	−0.07 (−0.20–0.05)	0.256
Fat mass (kg)	0.02 (−0.09–0.13)	0.710
**Hormonal factors**		
Testosterone (nmol/L)	0.07 (−0.02–0.18)	0.133
DHEA (nmol/L)	0.15 (0.01–0.31)	0.042
Cortisol (nmol/L)	0.003 (−0.002–0.008)	0.317
BDNF (nmol/L)	−0.07 (−0.21–0.05)	0.257
CRP (nmol/L)	−0.02 (−0.21–0.13)	0.840
IGF−1 (nmol/L)	0.07 (−0.05–0.19)	0.241
SHBG (nmol/L)	−0.02 (−0.08–0.03)	0.393
**Psychological factors**		
Hardiness	−1.03 (−2.83–0.57)	0.226
Sense of coherence	−0.25 (−1.02–0.51)	0.525
Depression	0.43 (−0.88–1.70)	0.496
Stress	0.54 (−0.56–1.66)	0.334
Cohesion	−0.49 (−1.88–0.83)	0.471
Big Five: neuroticism	0.14 (−0.45–0.72)	0.633
Big Five: extraversion	−0.28 (−0.86–0.27)	0.313
Big Five: openness	−0.38 (−1.01–0.22)	0.215
Big Five: agreeableness	−0.49 (−1.30–0.26)	0.204
Big Five: conscientiousness	−0.33 (−0.97–0.29)	0.294
Adverse childhood experiences	−0.07 (−0.33–0.20)	0.590
Core self-evaluation: self-esteem	0.52 (−0.64–1.77)	0.388
Core self-evaluation: generalized self-efficacy	0.68 (−1.30–2.71)	0.499
Core self-evaluation: locus of control	0.10 (−1.08–0.84)	0.828
Core self-evaluation: neuroticism	−0.02 (−0.63–0.60)	0.954

All regressions adjusted for treatment condition beginning at day 6 of the military field training for treatment group.

**Table 3 ijerph-17-09064-t003:** Prediction model for dropout (Cox regressions with proportional hazard ratios) after using STEP algorithm among set of variables separately for physical fitness variables alone, body composition variables alone, hormonal variables alone and psychological factors alone.

Characteristics	Beta-Coefficients + 95% CI	*p*-Value
**Physical fitness**		
12 min running test (m)	0.998 (0.996–1.00002)	0.052
Maximal isometric force of the lower extremities (kN)	0.0996 (0.0991–0.1001)	0.129
**Body composition**		
Body height (cm)	0.08 (−0.04–0.21)	0.226
% Body fat	0.04 (−0.07–0.16)	0.435
Skeletal muscle mass (kg)	−0.14 (−0.33–0.02)	0.103
**Hormonal factors**		
Cortisol (nmol/L)	1.006 (1.0003–1.01)	0.039
IGF-1 (nmol/L)	1.09 (0.98–1.21)	0.096
**Psychological factors**		
Hardiness	0.29 (0.06–1.30)	0.105
Depression	0.10 (0.009–1.17)	0.066
Stress	2.95 (0.65–13.32)	0.159

All regressions adjusted for treatment condition beginning at day 6 of the military field training for treatment group.

**Table 4 ijerph-17-09064-t004:** Prediction model for dropout (Cox proportional hazards regressions) including all variables in the same model after using STEP algorithm to reduce variables that are not expected to improve predictions.

Characteristics	Beta-Coefficients + 95% CI	*p*-Value
12 min running test (m)	0.997 (0.994–0.999)	0.006
Cortisol (nmol/L)	1.006 (1.001–1.011)	0.017
IGF-1 (nmol/L)	1.082 (0.994–1.177)	0.069

All regressions adjusted for treatment condition beginning at day 6 of the military field training for treatment group.

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
