# Peer review of "Can Physiological and Psychological Factors Predict Dropout from Intense 10-Day Winter Military Survival Training?"

_ijerph, 2020, doi:10.3390/ijerph17239064_

Round 1
Reviewer 1 Report
Dear authors
Although that your article has some limitations, it is an interesting research.
I believe that with a few corrections, the article will be able to be published.
You can see the review comments in the text o the article

Author Response
We want to thank the reviewer for insightful comments that improved the manuscript. Here are point by point responses.
Dear authors
Although that your article has some limitations, it is an interesting research.
I believe that with a few corrections, the article will be able to be published.
You can see the review comments in the text o the article
Response: Thank you for the positive feedback.
line 17: the wording of the purpose of the research is proposed to be the same as that in the introduction chapter
Response: Thank you. We have changed the purpose in the end of the introduction accordingly.
line 99: look at the comment in the abstract
Response: Thank you. We have revised accordingly.
line 114; write more information about your sample (how old were they on average). Also consider whether your sample exercised in their daily life or had more to do with sports.
Response: Thank you for the comment. We have added mean+sd age of the participants. In addition age is reported in table 1 for completers and dropouts separately. Majority of the participants exercised at least 2-5 hours a week (46%) or at more than 5 hours a week (48%). This has been added in the manuscript (lines: 113-115)
line114; Generally what were the selection criteria of the sample?
Response: The selection criteria for the sample was based firstly on eligible to participate to the survival training (withdrawal based only on medical doctor`s decision either on sickness or injury) and secondly on voluntariness to participate to the study. All conscripts were from the same unit and all of them were eligible and voluntary to participate to the study. We have added this information in the lines: 115-118.
line 139: Was it necessary to check the validity of the questionnaire? if yes, how did you check it and what was the value of the validity ?
Response: We used the NASA TLX questionnaire because it is widely used survey in different occupations and in tactical athletes nad also in other occupations (https://pubmed.ncbi.nlm.nih.gov/?term=nasa-tlx). The reliablility and validity of the NASA TLX questionnaire has been reported in their website (https://ext.eurocontrol.int/ehp/?q=node/1583):
Reliability:
The re-test reliability, split-half reliability, Cronbach's alpha coefficient and correlation coefficients between item score and total score were adopted to test the reliability:
-The re-test reliability coefficients varied from 0.516 to 0.753 (P < 0.01), indicating good re-test reliability.
- The split-half reliability of NASA-TLX and Cronbach's alpha coefficient were more than 0.80, the correlation coefficients between its items score and total score were all more than 0.60 (P < 0.01) except the item of performance.
-Both scales had good inner consistency.
- The Pearson correlation coefficient between the two scales was 0.492 (P < 0.01), implying the results of the two scales had good consistency.
Validity:
- The test of validity included structure validity Factor analysis showed that the two scales had good structure validity and it resulted extensively validated.
line 442: It is important to write the sample selection criteria in the methodology chapter. This will convince the reader of your article of the validity of the sample:
Response: Thank you for the comment. We have revised accordingly (lines: 113-115).
Reviewer 2 Report
This study on the variables predicting abandonment in military training is interesting and easy to read but could have been carried out with a better methodology and the following aspects must been improved:
Introduction
Line 44, 49-50, 71: On what do you base this statement?
Line 49-50: On what do you base this statement?
Line 54: Physical fitness defined in what way? previously you talked about the importance of psychology, now a better physical fitness would improve the results? it's a very risky affirmation.
Line 71: On what do you base this statement?
The last paragraph of the introduction includes aspects (line 104-106) that should not be in that place.
Materials and Methods
A drawing explaining in a simpler way what each group has done and when they have taken the data or made the assessments would be appreciated. It is very confusing.
Line 166: why do the references show up as underlined?
Line 177: Which resistance was selected for the test?
Line 179: Why did you use the horizontal jump and not the vertical one?
Line 250: why do the words show up as underlined?
Discussion:
You yourselves say that this type of training is very demanding physiologically and psychologically, so it can be assumed that hormone levels are affected. Can there be any relationship between better aerobic fitness and a different hormonal response in humans?
Line 350: Strength and power may have had nothing to do with desertion, but don't you think it could be because of the types of tests used?
There is a poor discussion as to why some measured fitness variables are unrelated to dropout.
Line 374: The cortisol data recorded are high compared to those recorded in other studies? even if they are not from the same studies as yours.
Line 388: Why do you think that anthropometry and body composition are not related to desertion?
Line 393: Why do you think there was no relationship?
Limitations:
The tests used could also be a limitation. The methodology employed is also a limitation.
Author Response
We want to thank the reviewer for insightful comments. We think the manuscript improved a lot.
This study on the variables predicting abandonment in military training is interesting and easy to read but could have been carried out with a better methodology and the following aspects must been improved:
Introduction
Line 44, 49-50, 71: On what do you base this statement?
Response: Thank you for the comment. In Finnish Defence Forces, this is how we formulate our training for conscripts. Theory and first-stage learning of military skills are firstly done in garrison conditions. The second phase is to apply these skills in real-life conditions in the field. We find hard to find any reference to back-up this statement but rather we rely here on subject matter expertise. Therefore, we have not modified these sentences. If there are any specific suggestion or needed modifications by the reviewer we are naturally willing to revise.
Line 49-50: On what do you base this statement?
Response: Thank you for the comment. We have now revised this sentence and added references:
“Therefore, field training often includes high demands for both psychological and physiological tolerance due to energy deficit, sleep restriction and high amount of physical activity combined with demands for executive function (Nindl et al. 2007, Kyröläinen et al. 2008, Harris et al. 2005, Duman & Monteggia, 2006).”
Line 71: On what do you base this statement?
Response: Thank you for the comment. We have now modified this sentence with more robust focus and added references directed to decision making (Harris & Morgan 2005) and coping (Eid & Morgan 2006) as examples.
Line 54: Physical fitness defined in what way? previously you talked about the importance of psychology, now a better physical fitness would improve the results? it's a very risky affirmation.
Response: Thank you for the comment. We do agree. We have deleted this sentence.
The last paragraph of the introduction includes aspects (line 104-106) that should not be in that place.
Response: Thank you for the comment. We have now deleted this sentence here.
Materials and Methods
A drawing explaining in a simpler way what each group has done and when they have taken the data or made the assessments would be appreciated. It is very confusing.
Response: Thank you for the excellent comment. We do fully agree. As this study is part of the bigger study program it might be difficult for reader to understand the design (especially because intervention is included in the design, although those participants are intentionally censored after 5 days). Therefore, we have now drawn a flow chart (figure 1) to better elucidate timeline, participants, dropouts and the intervention and also specified the determination of dropout in logistic regressions
Line 166: why do the references show up as underlined?
Response: This is purely a mistake. We have corrected it.
Line 177: Which resistance was selected for the test?
Response: Thank you for the comment. The resistance depended upon the participants`body weight varying between 61-95 kg. More details can be found from the manufacturer website or a previous study:
https://support.wattbike.com/hc/en-gb/articles/360013621359-6-second-test-Recommended-resistance-settings-
https://www.researchgate.net/publication/305222885_Validation_of_a_6_Second_Cycle_Test_for_the_Determination_of_Peak_Power_Output_PPO_Using_Wattbike_Cycle_Ergometer
We have now revised these sentences in the manuscript as shown below:
" A 6‐s maximal anaerobic power cycle ergometer test (Wattbike Ltd) was used to measure peak power in the lower extremities. The participants were seated stationary at the start with the dominant leg initiating the first down‐stroke. The weight of the participants was inserted into the test computer and air and magnetic resistance were set according to body weight with variance of options 3-5 (61-95 kg). The test started following a 5‐s countdown followed by verbal command. (Herbert, Sculthorpe, Baker, & Grace, 2015)."
Line 179: Why did you use the horizontal jump and not the vertical one?
Response: The horizontal jump is a part of the standard physical fitness testing battery in Finnish Defence Forces including both soldiers and conscripts. Therefore, the test was familiar to the participants and no separate familiarization was necessary.
Line 250: why do the words show up as underlined?
Response: This is purely a mistake. We have corrected it.
Discussion:
You yourselves say that this type of training is very demanding physiologically and psychologically, so it can be assumed that hormone levels are affected. Can there be any relationship between better aerobic fitness and a different hormonal response in humans?
Response: Thank you for an excellent comment. To our understanding there is at least weak link between aerobic fitness and some hormones in resting conditions (see our previous study focused on IGF-1 and fitness, PMID:21459641). Regarding the present study, it may be speculated that changes in fitness and hormones during the survival training could have been predictive of dropout. However, this remains topic for future studies.
Line 350: Strength and power may have had nothing to do with desertion, but don't you think it could be because of the types of tests used?
Response: Thank you for the comment. The results show consistently no association/predictive power to muscular fitness test results. We measured muscular fitness extensively (maximal strength for both lower and upper body, explosive strength for lower and upper body and also 1-min muscular endurance for core and upper body). In addition, the tests included both isometric and dynamic tests. Therefore we feel confident about null-association because we measured muscular fitness in multiple ways.
There is a poor discussion as to why some measured fitness variables are unrelated to dropout.
Response: Thank you for a critical comment. There are not many studies that have studied this issue previously. Therefore, comparing the results to other studies are quite limited. The mechanisms and underlying factors leading to either significant or non-significant predictions should of course be done. We have tried to improve the discussion as the best of our capability. Our discussion was based on the fatigue accumulation during the survival training, which most likely (based on the at least these findings) could be better counteracted with aerobic than muscular fitness (lines: 361-376) and thereby showing significant prediction be aerobic fitness. Further, we have also mentioned the importance of muscular fitness in broader view within soldiers task specific environment. If there appear any specific issue in discussion that the reviewer thinks is lacking we are happy to get feedback and improve the discussion.
Line 374: The cortisol data recorded are high compared to those recorded in other studies? even if they are not from the same studies as yours.
Response: Thank you for noticing this. We had errors in the table, where unit was wrong (should be nmol/L). We have now corrected this. The cortisol values are within the normal range in our sample size measured in our laboratory and are well in line with our previous studies (PMID: 18040709, PMID:32150971, PMID: 28972598). We understand that between different methods and laboratories there may exist differences in hormone concentration levels. Therefore, the comparison of levels of hormones between studies may be questionable.
Line 388: Why do you think that anthropometry and body composition are not related to desertion?
Response: We have extended our discussion about body composition with the following: " It may appear that narrow distribution of body composition variables within the study sample may affected the results towards non-significant findings. Furthermore, it may be that the ones with high body fat percentage or low skeletal muscle mass, both regarded as negative for soldiers` physical performance, may have had good aerobic fitness and therefore this masks predictive capacity with body composition variables, as have been shown previously in US enlistees (Niebuhr et al. 2009)."
Line 393: Why do you think there was no relationship?
Response: We think that psychological metrics at baseline may not be sensitive enough to predict dropout, based on the present study. We further think that changes in these variables across the survival training could be more sensitive markers for dropout prediction, as we have proposed for eg. hardiness, but that remains a topic future investigations. Also partly controversial results between the present and previous studies may be due to different study population (conscripts vs. Special Operators).
Limitations:
The tests used could also be a limitation. The methodology employed is also a limitation.
Response: We do agree. To assess physical fitness there are always different approaches and chosen methods. The strength of the present study is however, extensive physical fitness assessments including most physical fitness dimensions. Regarding methodology, we acknowledge that all studies have their limitations for example regarding design whether they are cross sectional, prospective or RCT, chosen statistical approach etc. We believe that within the present study design, we did the best we can and readers can justify the results according these known limitations.
Reviewer 3 Report
It is an original and innovative work that will allow military teams to try to avoid drop outs in the training field in winter, in cold environments. It is relevant because not having drop outs could be a determining factor in the battlefield.
In Table 1, I understand that there are no significant differences between the two groups. If there were a difference, it should be indicated.
In figure 1, for the 4 graphs, it would be interesting to show if one day is statistically different from another.
In lines 153-154 and 206-207, it is indicated that the basal fitness and psychological measures were measured 3 days before the study. However, it is indicated that the 12-minute test was done during the second week of military service. In these paragraphs it should be indicated how much time passed from the 12-minute test to field training, because one of the conclusions of the work is that the test could be relevant in selecting the military...
Finally, was there a questionnaire included for those military personnel who would be discharged? It would have been interesting to collect information on why they decided to leave training...
I encourage you to continue researching how to intervene on those who may be most likely to cause drop out, so that in the future, 100% of the military will be able to complete intensive field training.
Author Response
We want to thank the reviewer for reviewing this manuscript.
It is an original and innovative work that will allow military teams to try to avoid drop outs in the training field in winter, in cold environments. It is relevant because not having drop outs could be a determining factor in the battlefield.
Response: Thank you for the positive comments. We do also believe this study adds up an interesting and highly valuable information.
In Table 1, I understand that there are no significant differences between the two groups. If there were a difference, it should be indicated.
Response: Thank you. As we have done more sophisticated statistical analysis (Cox regressions) we decided not to do statistical analysis between these descriptive results in two groups. We followed the example of high standard publications (eg. Lancet), where descriptive results are not allowed to be compared.
In figure 1, for the 4 graphs, it would be interesting to show if one day is statistically different from another.
Response: Thank for the comment. This is indeed interesting. However, we are about to study these differences together with some other outcomes in the future manuscript in more details. Therefore, we would like to leave this for later and rather here describe and show the overall demands over the course of the survival training without statistical analysis.
In lines 153-154 and 206-207, it is indicated that the basal fitness and psychological measures were measured 3 days before the study. However, it is indicated that the 12-minute test was done during the second week of military service. In these paragraphs it should be indicated how much time passed from the 12-minute test to field training, because one of the conclusions of the work is that the test could be relevant in selecting the military...
Response: Thank you. We have revised accordingly:
"Aerobic fitness was measured with a 12-minute running test (Cooper 1968) on outdoor track in the beginning of the military service (~ 2 or 8 months prior the beginning of the study depending on each conscript`s service time)"
Finally, was there a questionnaire included for those military personnel who would be discharged? It would have been interesting to collect information on why they decided to leave training...
Response: Unfortunately, we did not assess any questionnaires for discharged but that could be a topic for future studies.
I encourage you to continue researching how to intervene on those who may be most likely to cause drop out, so that in the future, 100% of the military will be able to complete intensive field training.
Response: Thank you for the positive comment. We do also believe that this area of research is too less studied and we hope to follow conducting future studies in this topic.
Round 2
Reviewer 2 Report
Thank you very much for the explanations and corrections provided.
Congratulations for the research.